# Benefits and Challenges of Inhibiting EZH2 in Malignant Pleural Mesothelioma

**DOI:** 10.3390/cancers15051537

**Published:** 2023-02-28

**Authors:** MHD Ouis Al Khatib, Giulia Pinton, Laura Moro, Chiara Porta

**Affiliations:** 1Department of Pharmaceutical Sciences, Università del Piemonte Orientale “Amedeo Avogadro”, 28100 Novara, Italy; 2Center for Translational Research on Autoimmune & Allergic Diseases (CAAD), Università del Piemonte Orientale “Amedeo Avogadro”, 28100 Novara, Italy

**Keywords:** EZH2, malignant pleural mesothelioma, epigenetic, tumor microenvironment, macrophages, immune infiltrate, immunotherapy, targeted therapy

## Abstract

**Simple Summary:**

Malignant pleural mesothelioma (MPM) is an aggressive cancer linked to asbestos exposure with an extremely poor outcome. Despite the recent approval of immune checkpoint blockade-based therapies, MPM still remains a fatal cancer that challenges physicians and scientists. Enhancer of zeste homolog 2 (EZH2) has emerged as a promising therapeutic target. In addition to being an oncogenic driver, EZH2-dependent epigenetic reprogramming modulates tumor-immune infiltrate. Therefore, we argue that a better understanding of the molecular mechanisms that sensitize cancer cells to EZH2 inhibition and modulate tumor microenvironment will likely provide important insights for new treatment options for MPM.

**Abstract:**

Malignant pleural mesothelioma (MPM) is an aggressive thoracic cancer that is mainly associated with prior exposure to asbestos fibers. Despite being a rare cancer, its global rate is increasing and the prognosis remains extremely poor. Over the last two decades, despite the constant research of new therapeutic options, the combination chemotherapy with cisplatin and pemetrexed has remained the only first-line therapy for MPM. The recent approval of immune checkpoint blockade (ICB)-based immunotherapy has opened new promising avenues of research. However, MPM is still a fatal cancer with no effective treatments. Enhancer of zeste homolog 2 (EZH2) is a histone methyl transferase that exerts pro-oncogenic and immunomodulatory activities in a variety of tumors. Accordingly, a growing number of studies indicate that EZH2 is also an oncogenic driver in MPM, but its effects on tumor microenvironments are still largely unexplored. This review describes the state-of-the-art of EZH2 in MPM biology and discusses its potential use both as a diagnostic and therapeutic target. We highlight current gaps of knowledge, the filling of which will likely favor the entry of EZH2 inhibitors within the treatment options for MPM patients.

## 1. Introduction

Malignant pleural mesothelioma (MPM) is an aggressive thoracic cancer that derives from the mesothelial cells of the pleura and is mainly associated with prior exposure to asbestos fibers [1]. Even though it is a rare cancer, the global rate of MPM is increasing because of the constant use of asbestos in some countries and the difficulty in its removing from the environment, even in countries that banned its use in the 1990s [2]. MPM has long been classified in three main histological subtypes, which are characterized by different frequencies and prognoses [3]. Specifically, epithelioid MPM represents the most common (50–70%) and the less aggressive subtype; sarcomatoid MPM is the rarest (10–20%), most aggressive and chemo-resistant subtype; and biphasic MPM is characterized by epithelial and mesenchymal components and is the subtype whose frequency and outcome are in between the previous ones. In addition to histology, both their stromal and molecular features are increasingly recognized as important prognostic determinants and are included in the updated classification of pleural tumors published by the World Health Organization (WHO) in 2021 [4]. Due to the long latency of tumor development—which usually takes approximately 40 years—and the poor specificity of the clinical symptoms, MPM is usually diagnosed in old individuals at an advanced stage, when malignant cells have already spread to all the pleural layers [2]. Therefore, MPM clinical management is challenging, and the high resistance of malignant cells to treatments further worsens the patient’s outcome. Overall, this results in a dismal prognosis and a-5-year survival rate of approximately 10%. Despite its poor effectiveness, the combination chemotherapy with cisplatin and pemetrexed has remained the only first-line therapy for MPM for almost two decades [5]. The addition of bevacizumab to combination chemotherapy showed a two-month-survival improvement, but it didn’t receive any approval because of the increased frequency of severe adverse events [6]. In contrast, the combination of Tumor Treating Fields (TTFields)—which is a non-invasive approach based on the transcutaneous delivery of low-intensity alternating electric fields—with gold standard chemotherapy showed promising results in terms of safety and efficacy in a single-arm phase-II multicentric study. As a result, it was approved in 2019 by the Food and Drug Administration (FDA) as a front-line therapy for unresectable, locally advanced or metastatic MPM. Nevertheless, the lack of randomized evidence limits its entry into the clinical guidelines. Meanwhile, the success of immune checkpoint blockade (ICB)-based immunotherapy for the treatment of melanoma [7] has fostered its evaluation for other deadly cancers, such as MPM. Despite the disappointing results of the first trials, in 2021 the publication of the Checkmate 743 trial, which was a large randomized open-label phase III study, showed that the combination of ipilimumab (anti-CTLA-4) and nivolumab (anti-PD-1) in a frontline setting is more effective than standard chemotherapy. As a result, both the FDA and the European Medicines Agency (EMA) rapidly approved the combined ICB as a new first-line treatment option for unresectable MPM. Although this undoubtedly represents a breakthrough for MPM, many patients are still refractory or relapse after a few months of therapy. Thus, MPM is still a fatal cancer that urgently needs new treatment options. The identification of reliable biomarkers that enable researchers to anticipate the diagnosis at a “pre-invasive” stage is obviously a key step towards better clinical management. Equally important are new therapeutic strategies along with predictive biomarkers to guide clinical decision making toward the best treatment for each patient, which are very active fields of research that challenge physicians and scientists.

Enhancer of zeste homolog 2 (EZH2) is a well-known oncogenic driver in different malignancies, wherein it regulates gene expression in a PRC2-dependent and -independent manner [8]. Although EZH2 alone is enzymatically inactive, biochemical and structural studies have shown that in association with EED, SUZ12 and RbAp46/48, it becomes the catalytic subunit of polycomb repressive complex 2 (PRC2), which represses gene expression by the trimethylation of histone H3 on lysine K27 (H3K27me3) [8]. Although this epigenetic repression of genes plays a key role during tissue development and stem cell fate decision, its dysregulation can bring about the silencing of tumor suppressor genes and the promotion of carcinogenesis. Additionally, emerging studies have pointed out that EZH2 can promote the activation of key oncogenic programs through its direct interaction with transcription factors, such as NF-κB, estrogen and androgen receptors [9]. Collectively, the overexpression or gain-of-function mutations of EZH2 have been reported in a variety of solid and hematological cancers [8]. Accordingly, EZH2 has been recently introduced by the WHO as a diagnostic marker that enables the distinction of MPM from benign mesothelial proliferation [10]. Because of the association between its overexpression and a worse outcome [11,12,13], its inhibition has also been evaluated for new therapeutic perspectives. In models of MPM overexpressing EZH2 due to BRCA-1-associated protein 1 (BAP1) loss, pharmacological EZH2 inhibition showed significant anti-tumor activity [14]. However, treatment with tazemetostat, an EZH2 inhibitor that has recently been entered into the clinical treatment for epithelioid sarcoma [15], showed only a modest response rate in patients with relapsed or refractory BAP1-inactivated mutations [16]. Therefore, there is a need to better understand the mechanisms that sensitize cancer cells to EZH2 inhibitors, along with their effects on the tumor microenvironment (TME). Indeed, EZH2-dependent epigenetic reprograming has emerged as a crucial modulator of tumor-infiltrating immune cells in different types of malignancies, but it has never been fully explored in MPM. As a result, combining EZH2 inhibitors with other treatment approaches, including immunotherapy, is currently a hot topic of research in solid tumors [17] and might represent the next key challenge for the clinical management of MPM. Based on these premises, this manuscript reviews the current state-of-the-art of EZH2 in MPM pathogenesis, diagnosis and therapy. Specifically, we provide a comprehensive narrative synthesis of the evidence regarding the identification of EZH2 in the context of MPM, we critically describe its value as a diagnostic biomarker, and we discuss the pre-clinical and clinical studies that identify EZH2 as a new promising therapeutic target. We also highlight current gaps of knowledge and argue about the putative therapeutic perspectives of EZH2 inhibitors in combination with ICBs for MPM.

## 2. EZH2 in MPM

The mutational landscape that has emerged over the previous years has highlighted extensive genetic variation and gene expression deregulation both between and within MPM patients [18,19,20,21]. This molecular heterogeneity suggests the existence of a continuum of MPM clinical phenotypes whose understanding will remarkably improve MPM classification and prognostication. In addition to the mutated genes that characterized each tumor, most MPMs also harbor loss-of-function mutations or the genetic loss of a few tumor suppressor genes (*CDKN2A/2B*, *BAP1*, *NF2*, *TP53*, *LATS2* and *SETD2*), which likely plays a key role in neoplastic transformation [18,19]. The inactivation of such tumor driver genes is mainly due to chromosomal instability rather than point mutations. As a result of chromoplexy or chromothripsis, multiple chromosomal rearrangements and deletions are commonly observed in MPM cells [22,23]. In addition, tumor suppressor genes can be silenced by epigenetic modifications. Since 2009, it has been known that the expression of up to 11% of the genes in MPM cells are repressed by histone and DNA methy24 is nowlation [24]. With the exception of some overlaps, the majority of the genes enriched with H3K27me3 have no detectable level of DNA hypermethylation on the CpG promoters, while most of the DNA hypermethylated genes have no H3K27me3 marks. Thus, it appears that H3K27me3 and DNA hypermethylation may contribute to MPM development through the silencing of specific target genes. Two years later, Kemp C.D. et al. provided the first evidence of the aberrant expression of the polycomb group (PcG) proteins in MPM and proposed its targeting as a new potential treatment for this malignancy [12]. They revealed that the majority of MPM cell lines and primary MPM cells express higher levels of EZH2—a core component of PRC-2—than normal mesothelial cells. More importantly, the immunohistochemical (IHC) analysis of the MPM specimens demonstrated that EZH2 overexpression was associated with aggressiveness and the advanced stage of disease, and it decreased patient survival. Albeit poorly studied in the context of pleural mesothelial cells [25], it is widely recognized that the balanced activity of EZH2 methyltransferase with KDM6A (UTX) and KDM6B (JMJD3) demethylase controls the physiological levels of H3K27me3, which drives proper cell differentiation during development. Accordingly, an accumulating amount of evidence has indicated that the dysregulated activity of these proteins is linked with cancer cell features (e.g., proliferation, survival, stemness, migration, epithelial-mesenchymal transition) in different tumor types [26]. However, the interplay between EZH2 methyltransferase and KDM6A (UTX) and KDM6B (JMJD3) demethylase has yet to be explored in MPM. The analysis of surgical samples from MPM patients showed that both KDM6A and KDM6B transcript levels were increased in malignant tumors [27]. However, their pharmacological inhibition resulted in stronger anti-proliferative effects in normal mesothelial compared to MPM-derived cell lines, reducing the interest in KDM proteins as therapeutic targets [27].

In contrast, the upregulation of EZH2 observed in tumor tissue biopsies were retained in the MPM-derived cell lines, suggesting that EZH2 expression is under the control of tumor-specific factors. Specifically, the expression of a number of PcG genes, including EZH2, is transcriptionally regulated by E2F1. Nevertheless, this control can be dysregulated in MPM due to frequent *CDKN2A* deletions or epigenetic modulation [28]. Loss of BAP-1, which is another common oncogenic driver in MPM, was also found to be associated with EZH2 upregulation in human MPM cell lines [14]. Additionally, epigenetic regulators such as microRNA (miR)-101 and miR-26a, which are down-regulated in primary MPM, negatively affect the expression of EZH2 [12]. Recently, we have demonstrated that the silencing or inhibition of SIRT1 in MPM cells induces EZH2 protein acetylation and stability, as well as augmented H3K27me3 levels [28]. 

Over the last years, the analysis of TCGA data has confirmed that EZH2 mRNA is highly expressed in MPM and is significantly associated with decreased survival [11]. Along the same line, the analysis of transcriptomic datasets of MPM by bioinformatic tools, which allows for the prediction of protein–protein interaction networks (PPIs), has recently identified EZH2 as well as Hyaluronan Mediated Motility Receptor (HMMR) as “core” genes of MPM development, progression and outcome [29]. In agreement with in silico analysis, the role of PRC2-dependent gene expression in MPM pathogenesis has been strengthened by different in vitro studies [11]. Having corroborated the observation that a subset of genes repressed in MPM exhibits H3K27me3 without DNA hypermethylation, McLoughlin K.C. and co-workers used microarray, qRT-PCR, immunoblot and immunofluorescence techniques to examine PcG gene/protein expression in a panel of MPM cell lines and normal mesothelial cells. The results demonstrated that the overexpression of EZH2 and, to a lesser extent, EED and SUZ12 is associated with the increase of H3K27me3 in approximately 80% of primary MPMs. EZH2 or EED knock-down by shRNA decreased global H3K27me3 levels and significantly inhibited the proliferation, migration, clonogenicity and tumorigenicity of MPM cells [11]. BAP1 loss has been found functionally linked with EZH2 overexpression. Data obtained by LaFave L.M. et al. suggested that BAP1 interacted and co-occupied the EZH2 promoter with L3MBTL2, a protein that binds E-box motifs and maintains H4K20me1. BAP1 loss led to reduced L3MBTL2 stability and increased EZH2 transcription. Therefore, the silencing or pharmacological inhibition of EZH2 has been reported to induce apoptosis in BAP1-mutant MPM cell lines and reduce their growth when subcutaneously injected in mice [14]. Recently, we have reported that low SIRT1 sensitized MPM cells to EZH2 inhibition, which significantly reduced MPM cell proliferation in vitro by arresting cells in the G0/G1 phase and inducing a senescent phenotype [28]. 

Collectively, these studies indicate that despite the existence of different mechanisms leading to EZH2 overexpression, this epigenetic regulator is a central orchestrator of MPM pathogenesis. Therefore, EZH2 might represent both a reliable diagnostic marker of malignancy and a novel target for the development of new therapeutic interventions. 

## 3. EZH2 Is a Novel Diagnostic Biomarker for MPM

Currently, MPM is primarily diagnosed with imaging procedures, followed by the immunophenotyping of paraffin-embedded sections from thoracoscopic biopsies or, in some cases, of cells recovered from pleural effusion samples [30]. In addition to cytological/histological analyses, molecular markers are essential to differentiate MPM from either metastatic adenocarcinoma [31] or reactive mesothelial hyperplasia (RMH) [32]. Despite being a benign process, RMH cytologically resembles epithelioid MPM, which is the most common and diverse subtype in terms of cytological and architectural complexity [33]. Moreover, in the attempt to advance both the diagnosis and prognosis of MPM, a growing number of researchers have focused their attention on the identification of a reliable panel of biomarkers for distinguishing mesothelial tumors at the “pre-invasive” stage from those that have already infiltrated the pleural layers. These studies will likely pave the way for earlier therapeutic interventions, which might also be more effective. According to recent International Mesothelioma Interest Group (IMIG) guidelines [34], the homozygous deletion of the 9p21 locus detected by fluorescence in situ hybridization (FISH) and/or BAP1 loss detected by IHC [34,35,36,37] are the most accurate biomarkers for distinguishing malignant from benign mesothelial proliferations. Nevertheless, there are some concerns regarding their clinical use. Regarding FISH analysis of the 9p21 locus, it is hard to define an appropriate cutoff to differentiate homozygous from hemizygous deletions. Additionally, FISH is an expensive and time-consuming technique that cannot be performed in every facility. Interestingly, Girolami I. et al. [38] have recently reported high concordance between 9p21 homozygous deletion by FISH and methylthioadenosine phosphorylase (MTAP) loss by IHC. Thus, the latter could represent a reliable option for detecting 9p21 deletion in a low-resource setting. MTAP might also be useful in combination with BAP1 to improve MPM diagnosis. Although the number of studies was insufficient to perform a pooled analysis, [39,40,41] it seems that a lack of MTAP and BAP1 has a higher sensitivity than BAP1 loss only.

To distinguish MPM from RMH, additional IHC markers such as desmin, epithelial membrane antigen (EMA), insulin-like growth factor mRNA binding protein 3 (IMP3), glucose transporter-1 (GLUT-1) and CD146 have also been evaluated [42,43]. Yoshimura et al. reported that GLUT1 (up to 89%) and IMP3 (up to 94%) have the highest sensitivity, while Sheffield et al. found that EMA with p53 (64%) and BAP1 with 9p21 locus (100%) are the most sensitive and specific combinations, respectively. 

A recent systematic literature review confirmed that, unless they are used in combination, biomarkers such as GLUT1 and IMP3 have an unsatisfactory diagnostic performance [38]. 

Given that different studies have reported that EZH2 is overexpressed in a remarkable number of MPM cases (44.4–57%) but not RMH cases, EZH2 has emerged as an interesting diagnostic marker [42,44,45]. Indeed, EZH2, which is known to be upregulated in different solid cancers, is not a tissue-specific marker of malignancy. Therefore, high EZH2 expression can be exploited to distinguish MPM from benign mesothelial proliferations, but not from other lung malignancies. In contrast, there is evidence that BAP1 is a specific and useful marker for distinguishing non-mesothelial malignancies from epithelioid and biphasic but not sarcomatoid MPM in the thoracic or abdominal cavities. The latter rarely harbors BAP1 loss and is usually well-diagnosed on the bases of its histological features only. 

Even though enhanced EZH2 expression can be functionally associated with BAP1 loss in MPM cell lines [14], different studies have demonstrated that BAP1 loss is not statistically associated with EZH2 expression in human MPM biopsies [45], indicating that the mechanisms underlying EZH2 overexpression and BAP1 loss may be distinct. Thus, the combination of BAP1 and EZH2 detection by IHC could be a highly sensitive (90.0%) and specific (100%) approach for MPM diagnosis. Additionally, the lack of correlation among BAP1 or MTAP loss and EZH2 overexpression (*p* = 0.973, *p* = 0.284) suggests that the combination of the three different markers might further increase the accuracy of MPM diagnosis [42]. Recently, EZH2 has been evaluated in combination with Survivin, whose expression was detected in 67.9% of MPM cases, but not in RMH cases [44]. With the exception of some variations in terms of the prevalence of Survivin-positive MPMs across different cohorts of patients, [46,47], this study confirmed the diagnostic value of Survivin. Along the same line, the authors corroborated a highly significant direct association between BAP1 loss and Survivin expression [32], but also revealed an inverse association between high EZH2 expression and either BAP1 loss or Survivin expression. Therefore, the combinations of EZH2^high^ and/or BAP1 loss with Survivin^+^ might be exploited to gain sensitivity in the differential diagnosis between epithelioid MPM and RMH. 

It is worth noting that BAP1 and EZH2 are the only markers that are localized in the nuclei of tumor cells, whereases the IHC analysis of the other markers results in a cytoplasmic staining wherein variable intensity can challenge the detection of a positive signal from the background. Therefore, the inclusion of BAP1 and EZH2 in the panel of markers for the IHC analysis of tissue biopsies could greatly improve the accuracy of MPM diagnosis.

Previous systematic reviews have failed to define a reliable panel of diagnostic biomarkers for MPM. The variations in marker expression reported across the different studies may be reasonably assumed to be due to the differences in terms of sample sizes, antibodies used, staining and quantification techniques. Therefore, the standardization of IHC procedures will likely allow for the determination of the appropriate combination of markers that, together with histologic analysis and clinical evaluation, might anticipate the diagnosis of MPM.

## 4. EZH2 as a Promising Therapeutic Target for MPM

A growing number of studies have indicated the therapeutic potential of EZH2 targeting (Figure 1). The first evidence dates back to 2012, when 3-deazaneplanocin A (DZNep) demonstrated a significant cytotoxic effect against MPM cells [12]. DZNep is a S-adenosylhomocysteine hydrolase (SAH) inhibitor that indirectly inhibits EZH2 by interfering with S-adenosyl-methionine (SAM) and SAH metabolism. However, H3K27 demethylation observed upon DZNep treatment is due to the proteolytic degradation of EZH2 and other PRC2 components, rather than specific EZH2 catalytic inhibition [12]. In vitro, DZNep triggers the expression of several tumor suppressor genes, which inhibits MPM cell proliferation and induces cell senescence but not apoptosis [12]. These data are in accordance with p21^cip^ upregulation and the delay of the G2/M transition, which have been respectively observed in melanoma and breast cancer cells upon EZH2 knockdown [48,49]. Additionally, the effect of DZNep was evaluated on MPM xenografts. The results demonstrated a significant reduction of tumor size after each cycle of treatment and an approximately 50% decrease in tumor mass at the end of the treatment course, along with no signs of systemic toxicity. Therefore, the authors claimed that DZNep recapitulated, in vitro and in vivo, the effects of EZH2 or EED depletion in MPM cells.

Successively, LaFave L.M. et al. [14] proved that human *BAP1*-mutant MPM cell lines were sensitive to the selective EZH2 inhibitor EPZ011989. Accordingly, EPZ011989 significantly reduced the growth of sub-cutaneous transplanted *BAP1*-mutant MPM cells and abrogated pulmonary metastasis when mice were injected with a *BAP1*-mutant MPM cell line with metastatic potential. Because the wild-type tumors were less responsive to EZH2 inhibition, they concluded that *BAP1* mutations, which typically result in increased EZH2 expression, render MPM cells addicted to PRC-2. Despite the strong association between *BAP1* mutations and repression of PRC-2 targets [51], it seems that *BAP1* mutant MPMs harbor different clinical phenotypes, since different studies have reported an overexpression of EZH2 in *BAP1* wild-type MPM biopsies [28,42,45]. Recently, we have demonstrated that in low SIRT1 conditions, EZH2 inhibition significantly reduced the proliferation of *BAP1* wild-type MPM cells [28]. Interestingly, we have observed that EZH2 inhibition induced cell senescence by promoting *CDKN2A*/p16^ink4a^ expression, whereas *CDKN2A* null cells underwent apoptosis upon treatment with the EZH2 inhibitor EPZ6438 [28]. These findings indicate that patients carrying homozygous deletion or loss-of-function mutations of *CDKN2A* should be more responsive to EZH2 inhibition. Therefore, in a translational perspective, studies are warranted to evaluate *CDKN2A* status as a marker for patients’ stratification and/or potentiation of EZH2 inhibition efficacy.

A high-throughput screening (HTS) campaign followed by hit triaging led to the discovery of the EPZ005687 compound by the company Epizyme. This EZH2 inhibitor has a greater than 500-fold selectivity against 15 other protein methyltransferases and a 50-fold selectivity against the closely related enzyme EZH1 [52]. The EPZ005687 has a similar affinity for wild-type and Y641 mutant EZH2, but a greater affinity for the A677G mutant. In spite of the remarkable reduction of H3K27me3 in both EZH2 wild-type and mutant lymphoma cell lines, similar to other EZH2 inhibitors, EPZ005687 significantly inhibited the proliferation of mutant EZH2 cells only. A further-improved version of EPZ005687 is EPZ-6438 (tazemetostat), which is a potent and selective SAM competitive small molecule that retains the cellular activity and selectivity of EPZ005687 but gains better oral bioavailability and pharmacokinetic properties [53]. In addition to hematological malignancy, the inhibition of EZH2 can be beneficial for the treatment of solid cancers. Firstly, EPZ-6438 has demonstrated significant anti-tumor activity against malignant rhabdoid tumors (MRTs), provided that SMARCB1 is deleted. Indeed, EZH2 inhibition by EPZ-6438 induced apoptosis in SMARCB1-mutant MRT cells and dose-dependent tumor regression in xenograft-bearing mice [54]. Subsequently, accumulating preclinical studies have substantiated the therapeutic potential of EPZ-6438 for a variety of solid tumors [55], leading to the initiation of clinical trials worldwide. In 2020, tazemetostat (Tasverik^TM^) was approved by the FDA for the treatment of adults with locally advanced or metastatic epithelioid sarcoma not eligible for complete resection [15]. Along the same line, a phase-1 study recently conducted in Japan has reported that tazemetostat has a favorable safety profile and promising anti-tumor activity in patients with relapsed, refractory or advanced B-cell non-Hodgkin lymphoma [56]. However, some B-cell malignancies are resistant to EZH2 inhibitors [57], and in many solid cancers, despite the overexpression of EZH2, its inhibition alone doesn’t achieve a sufficient level of efficacy [58].

Consistent with that, the results of a recent multicenter single-arm open-label phase-II study with tazemetostat in *BAP1*-inactivated relapsed or refractory MPM patients provide the first evidence of safety along with a moderate anti-tumor activity [16]. The study enrolled patients with a more indolent disease after initial systemic therapy and included a substantial proportion of patients who had a surgical resection that did not reflect the real average of patients usually eligible for surgery. *BAP1* mutation was determined by DNA sequencing, while loss of protein expression was done by IHC. The primary end-point of the study was the disease control at 12 weeks. Indeed, meta-analyses of trials conducted in MPM patients indicates that this parameter is a reliable positive predictor of survival. The end-point was reached in about a half of the patients, and the drug showed a favorable safety and tolerability profile. Two patients had a partial response, with a 30-week median duration of response. Noteworthy, a preliminary exploration of the TME composition before and after treatment with tazemetostat highlighted a significant reduction of intra-tumoral and stromal B-cells. That effect on immune cells warrants future studies to gather its role on clinical response. 

Altogether, these findings indicate that tazemetostat is a promising therapeutic option, whose efficacy might likely be improved by a better definition of predictive biomarkers for the stratification of MPM patients, as well as by novel combination strategies of EZH2 inhibitors with therapies such as chemo-, immuno- and targeted therapy. Indeed, many preclinical studies have demonstrated the efficacy of EZH2 inhibition in combination with cisplatin in different tumor types, such as lung, ovarian, and breast cancers [59]. EZH2 inhibition can rescue cisplatin resistance and mitigate the adverse effects [59]. Given that cisplatin-based chemotherapy is the standard-of-care for MPM, future studies are warranted to evaluate the putative beneficial effects of the combination of EZH2 inhibitor with cisplatin.

## 5. The MPM Immune Microenvironment

In addition to cancer cells, the TME, including immune and non-immune cells, the extracellular matrix and the soluble mediators released by the different cells, plays a key role in MPM development, growth, progression and response to therapy [60,61,62]. Here, we focus on immune cells only (Figure 2), among which macrophages emerge as key orchestrators of both early tumor-promoting inflammation in response to asbestos fibers and immunosuppression at the advanced stage of MPM. Alveolar macrophages, which efficiently eliminate dust particles and environmental pollutants [63], struggle to clear fibers longer than 5 μm, which consequently remain in the lungs—triggering the neoplastic transformation of mesothelial cells. Although the underlying mechanisms have not yet been fully understood, it is widely recognized that ‘’frustrated phagocytosis’’ promotes a chronic inflammatory microenvironment that supports the carcinogenesis, survival and proliferation of neoplastic cells through the production of reactive oxygen and nitrogen species (ROS and NOS), as well as cytokines, such as IL-1β [64,65] and TNFα [66]. Additionally, High Mobility Group Box1 Protein (HMGB1), a damage-associated molecular pattern released by both mesothelial cells and macrophages, plays a key role in tumor development and progression by enhancing both macrophage-driven inflammation and mesothelial/neoplastic cell survival, proliferation, autophagy and epithelial-mesenchymal transition (EMT) [67,68]. Accordingly, HMGB1 dramatically increased in the blood of asbestos-exposed individuals, and its high levels in MPM patients are associated with a worse outcome [69,70]. Tumor-associated macrophages (TAMs), which are the most abundant population of immune cells in human MPM [61], largely stems from monocytes recruited by chemotactic factors like CCL2, which is produced abundantly by mesothelial cells exposed to asbestos [60,71]. As a result, CCL2 levels increased significantly both in the pleural effusion (PE) and in the blood of MPM patients, in particular at the advanced stage, supporting the central role of macrophages across all stages of MPM development [72]. Accordingly, the number of TAMs defined by the pan-macrophage marker CD68 was associated with worse outcomes in non-epithelioid MPM [71]. Similar to other tumor types, TAMs upregulate M2 markers like CD163 and CD206, indicating a shift of polarized activation toward the alternative (M2) immunosuppressive program. In agreement, a positive correlation between stromal CD68+ macrophages and immunosuppressive Tregs was observed in MPM specimens [73]. Additionally, pleural effusion is enriched in molecules, such as macrophage colony stimulating factor (M-CSF) [74], transforming growth factor β (TGF-β) [75] and prostaglandin E2 (PGE2) [76], which are released by tumor cells and drive immunosuppressive macrophage differentiation in vitro. In line with human evidence, the accumulation of immunosuppressive and tumor-promoting TAMs was also confirmed in different pre-clinical models of MPM [77,78], where their depletion and/or M1-reprograming rescued anti-tumor immunity [78], in particular in combination with anti-PD1/PD-L1 blockades [79]. Although these studies overall support the therapeutic value of the approaches that target macrophages, the increasing evidence of the intra- and inter-tumor heterogeneity of human MPM [20,80,81] points out the need for a better understanding of TME and its cross-talk with cancer cells. Even though they account for less than 10% of immune infiltrate, both polymorphonuclear (PMN) and monocytic (M-) myeloid-derived suppressor cells (MDSCs) exert different tumor-promoting activities that negatively affect MPM outcome [82]. Both subsets exert important immunosuppressive activities, as demonstrated by the inhibition of proliferation and cytotoxic activity of autologous human T lymphocytes [82]. Further supporting the therapeutic potential of targeting MDSCs, the neutralization of GM-CSF in a preclinical model of MPM inhibits the accumulation of tumor-infiltrating PMN-MDSC, boosting anti-tumor immunity [83].

Dendritic cells (DC), which play a key role in inducing an antigen-specific immune response, are not only reduced in number but also in migratory and antigen presentation capability. Although these cells maintain expression of IL-12, they also tend to produce higher amounts of anti-inflammatory and pro-angiogenic factors such as IL-10 and vascular endothelial growth factor (VEGF) [84]. 

So far, cytotoxic immune cell populations like NK and NKT cells have been poorly studied in human MPM. Different evidence indicates that, despite playing a relevant role in anti-tumor immunity, NK frequency in MPM is not associated with a better outcome [60]. A reasonable explanation is that the immunosuppressive microenvironment of MPM hampers their effector functions [85]. According to this hypothesis, in the PE of MPM patients, NK cells express high levels of the checkpoint inhibitors T-cell immunoglobulin and mucin-domain containing-3 (TIM-3) and lymphocyte activation gene-3 (LAG-3) [60]. Additionally, a reduced expression of activating receptor-like NKp46 and an enrichment of a CD56^Bright^ NK subset have been reported in the blood of MPM patients [86]. Interestingly, anti-CTLA-4-based immunotherapy seems to enhance the cytotoxic activity of NK cells since an increase of CD56^Dim^/CD56^Bright^ NK ratio has been observed in the blood of tremelimumab-treated patients [86]. NKT cells, whose activation by alpha-galactosylceramide in combination with cisplatin has demonstrated a relevant anti-tumoral activity in mouse models of MPM [87,88], represent an additional population of cytotoxic immune cells that warrants more study in MPM patients. 

Beyond the impact of each immune cell population, understanding the cross-talk among the stromal, immune and cancer cells is a key challenge for improving patients’ stratification and clinical management. Indeed, different studies based on the IHC analysis of immune infiltrate have observed that the combination of different immune cells has a better prognostic value than the frequency of single immune cell subsets. For example, although a high frequency of either T (CD3+, CD8+, or CD4+ cells) or B (CD20+ cells) lymphocytes has been reported as favorable prognostic markers [89,90,91,92,93] in epithelioid MPM, CD20^+^B cells^high^ CD163+ TAM^low^ and CD8+ T cells^low^ CD163+ TAM^high^ combinations showed a superior accuracy in predicting better and worse outcomes, respectively [92]. Additionally, in a cohort of patients with non-epithelioid MPM, it has been observed that despite the presence of a high number of anti-tumoral CD8+ T lymphocytes, when a significant level of CD68+ macrophages and PD-L1+ tumor cells are present as well, the response to chemotherapy and the outcome are poor [93]. In contrast, a higher number of B lymphocytes, along with the presence of tertiary lymphoid structures (TLS) consisting of B and T lymphocytes, have been associated with a response to chemotherapy and a longer survival for patients with epithelioid MPM [94]. These studies highlight the importance of TME composition not only as prognostic marker, but also as a predictor of response to therapy. Accordingly, a recent study performed on a small cohort of patients showed that a high number of CD8+ T cells is an independent factor associated with better survival in epithelioid MPMs treated with hypo-fractionated radiation therapy [95]. Besides chemo- and radiotherapy, the immune contexture obviously holds great promise as a predictor of response to immunotherapy. To overcome the limits of IHC, the development of innovative multiplex immunophenotyping techniques has marked a milestone for a more comprehensive characterization of the TME. Nevertheless, only Lee H. S. and colleagues have hitherto analyzed the MPM immune infiltrate by mass-cytometry [96]. As a result, MPM patients were stratified in two groups characterized by a distinct immunogenic immune signature, which was associated with favorable outcomes and a response to checkpoint blockade [96]. Although the multiplex immunophenotyping technique allows for the analysis of intratumor heterogeneity at the single-cell resolution level, transcriptional profiling is an easier approach that has become widespread over the last years. As a result, an underestimated level of cancer cell heterogeneity beyond histological subtypes has emerged. Additionally, due to the consistent increase of publicly available datasets, different algorithms have been generated to unravel the MPM microenvironment and determine the immune signatures to predict outcomes and response to treatments. For example, the application of the ESTIMATE algorithm has indicated a prognostic signature consisting of 14 stromal/immune-related genes, which could also be useful to predict response to ICB [97]. Recently, using non-negative matrix factorization (NMF) and nearest template prediction (NPT) algorithms, Yang and co-workers developed an in silico classification system that stratifies MPM in different immune subtypes that are associated with different prognoses [98]. In addition, because of the high lymphocyte infiltration, TCR and BCR diversity, and IFNγ signature, the “immune activated” subtype has a favorable response to ICB, while the “immune suppressed” subtype, which is characterized by a huge number of immunosuppressive Treg and myeloid cells (TAM, MDSC) along with a TGF-β signature, is resistant to ICB, but it could benefit from drug targeting macrophages such as CSF1/CSF1R antibody. Therefore, improving our understanding of the TME contexture prior to therapy could be crucial to guide clinical decision making, whereases gathering the effects of treatments on TME would provide a more comprehensive knowledge of their efficacy and might open new strategies to enhance their therapeutic effects. 

## 6. Effects of EZH2 Targeting on MPM Immune Infiltrate Are Still Largely Unknown

It has long been known that, besides the cancer cell-autonomous effect, the anti-cancer activity of drugs targeting epigenetic modulators is due to the promotion of anti-tumor immunity [99,100,101,102]. Although poorly studied in MPM, EZH2-dependent epigenetic reprograming can modulate tumor cell immunogenicity and TME composition, and it can directly regulate immune cell differentiation and functional activation (Figure 3). 

Specifically, in different types of both hematological (e.g., diffuse large B-cell lymphoma) and solid cancers (e.g., neuroblastoma, melanoma, breast, prostate and lung cancer), gain-of-function mutations or the overexpression of EZH2 increases H3K27me3, which represses genes encoding tumor-specific antigens and MHC molecules [103,104,105,106]. Therefore, EZH2 inhibitors can enhance tumor cell immunogenicity by reshaping the epigenetic landscape of cancer cells and favoring the expression of genes associated with both the presentation of new antigens and the recruitment of anti-tumor immune cells. Consistently, in preclinical models of ovarian cancer and melanoma, epigenetic reprogramming due to EZH2 knock down or pharmacological inhibition enhanced the expression of Th1-recruiting chemokines (e.g., CXCL9, CXCL10), increased tumor-infiltrated CD8+ T cells, and improved the efficacy of ICB-based immunotherapy [106,107]. 

Additionally, in a poorly immunogenic melanoma model, the inhibition of EZH2 triggered the expression of STING and consequently sensitized cancer cells to STING agonists. As a result, a combination of a EZH2 inhibitor and a STING agonist synergistically reduced tumor growth in association with an increased CD8+ T-cell infiltration [108]. Although the mechanism is different, the activation of STING upon treatment with EZH2 inhibitors has been also reported in prostate cancer. Indeed, in prostate cancer cells, EZH2 inhibitors can rescue the expression of endogenous retrovirus (LTR/ERV), which results in a “viral mimicry” state. Specifically, dsRNA molecules activate STING receptors, which triggers the expression of interferon-stimulated genes (ISGs). This brings about an increase of antigen presentation, cytotoxic CD8+ T cell recruitment and anti-PD1 responsiveness [109]. 

In line with these studies, using a MPM multicellular spheroid model (MCS), we have found that treatment with the EZH2 inhibitor tazemetostat lead to the upregulation of chemokines specific for the recruitment of cytotoxic immune cells such as CXCL9 and CXCL10 [50]. However, we have also found an increased expression of different monocyte chemoattractants (e.g., CCL2, M-CSF, CCL5, CXCL12, VEGF) in association with a significantly higher recruitment of tumor-promoting monocytes in the MCS [50]. This was the first study that had evaluated the effect of EZH2 on MPM TME composition, specifically on human monocytes and their impact on cancer cell responsiveness to tazemetostat. Subsequently, a functional association between EZH2 and TAM infiltration has been also reported in other types of tumors, such as breast and colorectal cancer (CRC) [110,111]. 

Recently, the effect of EZH2 on the composition of human MPM immune infiltrate has been explored using bioinformatic analysis on TCGA datasets. Interestingly, the results showed that high EZH2 expression, which is significantly associated with a worse outcome, negatively correlated with the number of tumor-infiltrating mast, NK and Th17 cells [13,60]. Overall, these studies provide the proof-of-concept that EZH2 modulates the composition of both innate and adaptive immune infiltrate in MPM.

Besides recruitment, EZH2 affects anti-tumor immunity by modulating the differentiation and functional activation of the immune cells [112]. Concerning T cells, EZH2 promotes the lineage-specification, identity, maintenance and survival of differentiated antigen-specific CD4+ T helper cells, whereas effector CD8+ T cell differentiation is restrained by EZH2, which favors the formation of precursor and mature memory CD8+ T cells [113]. Additionally, Treg differentiation and suppressive activity require the EZH2-dependent deposition of H3K27me3 marks [114,115]. Indeed, mice carrying Treg-specific Ezh2 deficiency showed a reduced growth of different types of tumors (e.g., CRC, melanoma, prostate cancer) in association with the reprograming of tumor-infiltrating Tregs in anti-tumor effector cells (e.g., IL-2, IFNγ, and TNF) [116]. Regarding innate lymphoid cells, EZH2 inhibits invariant natural killer T (iNKT) cell differentiation and function, as well as the maturation, activation, survival and cytotoxicity of NK cells [117]. Accordingly, in hepatic cancer, the inhibition of EZH2 in tumor cells enhanced NK recruitment via CXCL10 [118], and it enhanced their activation through the expression of NKG2D ligands [119]. EZH2 also modulates the differentiation of MDSC. In murine models of either CRC or Lewis lung cancer (LLC), blocking EZH2 with GSK126 in immunocompetent mice impaired anti-tumor immunity by boosting systemic MDSCs expansion and accumulation in TME. Depleting MDSCs by anti-GR1 neutralizing antibodies or low doses of gemcitabine/5-Fluorouracil rescued GSK126 efficacy by recovering the anti-tumor effector T-cell activity [120]. Divergent effects of EZH2 on TAM functional activation have been reported in different settings. In a murine model of MPM, it has been observed that the treatment of murine RAW264.7 macrophages with a EZH2 inhibitor led to the upregulation of the phagocytosis inhibitory checkpoint PD-1 and, consequently, impaired their cytotoxic activity toward the MPM cells in vitro and in vivo [121]. Accordingly, by using an MCS model consisting of human MPM cells and monocytes, we have demonstrated that tazemetostat enhances both the recruitment and M2-polarized activation of monocytes, blocking the anti-proliferative effects of EZH2 inhibition in cancer cells [50]. Therefore, combining EZH2 inhibition with TAM-targeted therapy, such as anti-CSF1R [122], might synergistically improve the anti-tumoral efficacy. Along the same line, the treatment of breast cancer cells with EZH2 inhibitors promotes recruitment and favors M2 polarized macrophage activation by inducing CCL2 upregulation [110]. In contrast, EZH2 depletion caused an miR-124-3p-dependent inhibition of CCL2 expression in the tumor cells, leading to the inhibition of M2 polarized activation [110]. This highlighted an additional level of complexity in EZH2 activity, whose non-enzymatic modulatory functions are still poorly characterized. Moreover, cancer cell intrinsic and TME signals may account for the distinct effects of EZH2 inhibitor in different tumor types. Indeed, in a murine colorectal cancer (CRC) model, tazemetostat induced the accumulation of anti-tumor macrophages [111]. Accordingly, in glioblastoma multiforme, EZH2 inhibition by DZNep favored macrophage M1 polarization, as demonstrated by the upregulation of pro-inflammatory cytokines and the downregulation of anti-inflammatory ones, and it enhanced phagocytic capability [123]. These divergent results suggest that cancer cell intrinsic and TME signals may account for the distinct effects of EZH2 inhibitor in different tumor types.

## 7. Conclusions

After decades of failed trials, the approval of immunotherapy based on the combination of ipilimumab and nivolumab has marked a milestone for MPM, particularly the sarcomatoid subtype, which is more aggressive and resistant to chemotherapy. However, MPM remains a deadly cancer with an unacceptably poor survival rate after diagnosis. Besides histology, the increasing advancements in MPM classification by molecular markers represent a key step towards better clinical management. Indeed, if we were able to bring the diagnosis toward the “pre-invasive stage” and to improve prediction of outcome, we would increase the chances of effective treatment regimens. In this context, EZH2 has emerged as a valuable diagnostic marker with a prognostic potential. Similar to many other solid cancers, its overexpression in MPM is recognized as an oncogenic driver. Consequently, inhibitors of EZH2 such as tazemetostat, which has recently entered into clinical use for epithelioid sarcoma, has attracted a lot of interest and has recently demonstrated some promising results of efficacy in preliminary clinical trials. Along with a better understanding of reliable biomarkers to identify the patients who most likely benefit from EZH2 inhibition, combinations of EZH2 inhibitors with different therapeutic modalities holds promise for enhancing efficacy. Being an epigenetic modulator, EZH2 has a profound effect not only on cancer cells, but also on TME. Given that EZH2 inhibitors can modulate both anti-tumor and pro-tumor immune cell populations, a better understanding of the effect of EZH2 inhibitors on the MPM immune infiltrate will likely help physicians determine the most effective combination approaches. Notably, the growing number of pre-clinical studies looking at different models of solid cancers indicate that EZH2 inhibitor synergizes with ICB-based immunotherapy thanks to the increased expression of PD-L1, immunogenic antigen and chemokine-recruiting cytotoxic T cells [124,125,126,127]. On the other hand, it is well-recognized that the efficacy of ICB-based immunotherapy could benefit by combination therapeutic strategies. So far, clinical trials conducted with MPM patients have evaluated ICBs with chemotherapy, targeted therapy like bevacizumab, and stereotactic body radiation therapy [128]. Epigenetic modulators, such as EZH2 inhibitors, which have been demonstrated to have a favorable safety profile along with a promising immunogenic potential, could represent a new potential therapeutic approach that warrants evaluation in combination with immunotherapy.

## Figures and Tables

**Figure 1 cancers-15-01537-f001:**
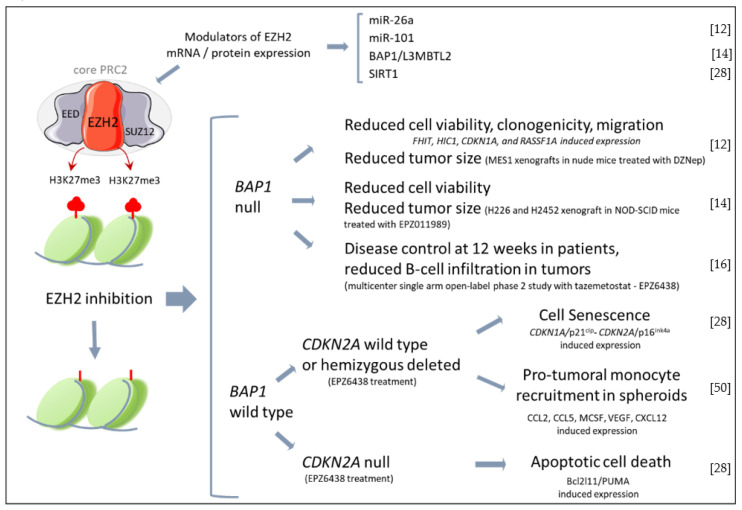
Effects of EZH2 inhibition in MPM. In the upper brace are reported miRNAs and proteins that have been described to modulate EZH2 expression in MPM cells. Below, the main effects of EZH2 inhibitors reported by in vitro and in vivo studies with MPM cells/tumors stratified by BAP1 or *CDKN2A* expression. (EZH2, enhancer of zeste homolog 2; EED, embryonic ectoderm development; PRC2, Polycomb re-pressor 2; SUZ12, suppressor of zeste 12 homolog; H3K27Me3, Histone 3 lysine 27 trimethylate; BAP1, BRCA1 associated protein 1; L3MBTL2, lethal 3 malignant brain tumor-like protein 2; SIRT1, sirtuin1; FHIT, Fragile Histidine Triad Diadenosine Triphosphatase; HIC1, HIC ZBTB Transcriptional Repressor 1, *CDKN1A*, cyclin dependent kinase inhibitor 1A; RASSF1A, Ras as-sociation domain family 1 isoform A; MCSF, macrophage colony stimulating factor; VEGF, vascular endothelial growth factor). The figure was partly generated using Servier Medical Art, provided by Servier, licensed under a Creative Commons Attribution 3.0 unported license https://creativecommons.org/licenses/by/3.0/ (accessed on 28 January 2023) [12,14,16,28,50].

**Figure 2 cancers-15-01537-f002:**
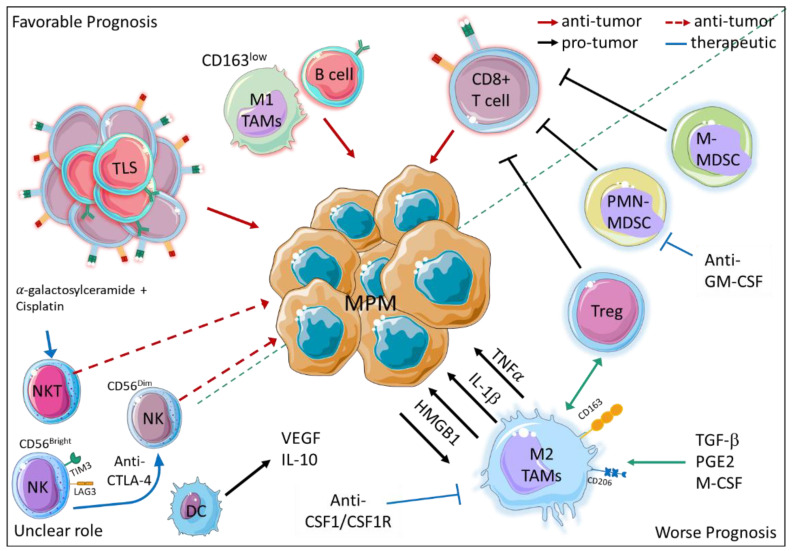
Immune cell infiltrate impacts MPM outcome. On the upper left corner, the main immune cell populations associated with better outcomes are depicted, whereas on the lower right corner, the pro-tumoral and immunosuppressive immune cell populations are shown. The immune cells with a putative albeit not yet proven anti-tumor activity are indicated in the lower left corner. (Solid red arrows: anti-tumor activity, dashed red arrows: putative anti-tumor activity, black arrows: pro-tumor activity, blue arrows and inhibition arrows: putative therapeutic approaches. TLS, tertiary lymphoid structure; TAM, tumor associated macrophage; PMN-MDSC, polymorphonuclear- myeloid-derived suppressor cells; M-MDSC monocytic-myeloid-derived suppressor cells; NK, Natural killer cells; NKT, Natural killer T cells; TAM, Tumor associated macrophages; DC, dendritic cells; Treg, T regulatory cells; IL-1β, Interleukin 1 Beta; IL 10, Interleukin 10; HMGB1, high mobility group box 1; PGE2, Prostaglandin 2; M-CSF, Macrophage colony stimulating factor; TNFα, Tumor necrosis factor alpha; TGF β, transforming growth factor beta; VEGF, vascular endothelial growth factor.) The figure was partly generated using Servier Medical Art, provided by Servier, licensed under a Creative Commons Attribution 3.0 unported license https://creativecommons.org/licenses/by/3.0/ (accessed on 28 January 2023).

**Figure 3 cancers-15-01537-f003:**
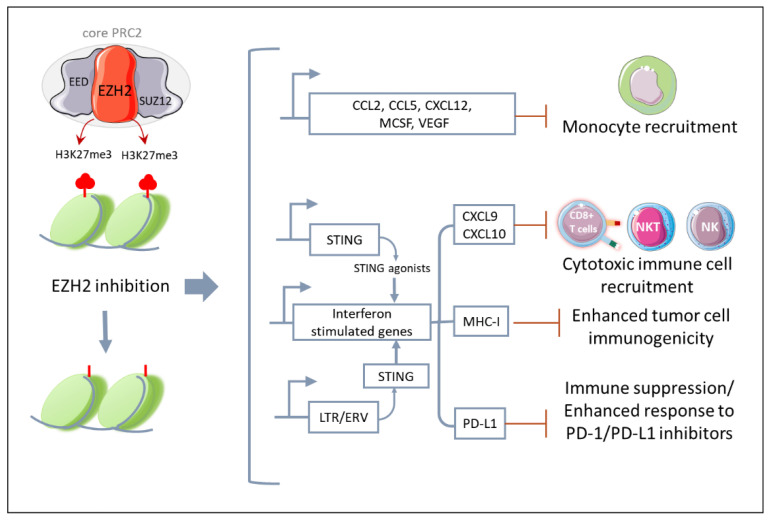
EZH2 modulates anti-tumor immunity. EZH2 inhibition leads to the epigenetic reprograming of cancer cells, which upregulates transcriptional programs associated with increased tumor cell immunogenicity and recruitment of cytotoxic immune effector cells, but also monocytes and immunosuppressive molecules such as PD-L1. This suggests that combinatory strategies targeting the tumor-infiltrating immune cells, such as anti-PD-1/PD-L1 antibodies, might synergize with EZH2. (STING, stimulator of interferon genes; CXCL9, CXC motif ligand chemokine ligand 9; CXCL10, CXC motif ligand chemokine ligand 10; MHC 1, Major histocompatibility complex 1; LTR, Long termina repeat; EVR, endogenous retrovirus; PD-L1, programmed death-ligand 1; PD-1, programmed cell death protein 1.) The figure was partly generated using Servier Medical Art, provided by Servier, licensed under a Creative Commons Attribution 3.0 unported license https://creativecommons.org/licenses/by/3.0/ (accessed on 28 January 2023).

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
