# Peer review of "Benefits and Challenges of Inhibiting EZH2 in Malignant Pleural Mesothelioma"

_cancers, 2023, doi:10.3390/cancers15051537_

Round 1

Reviewer 1 Report

The manuscript titled ‘Benefits and challenges of inhibiting EZH2 in malignant pleural mesothelioma’ is thoroughly a well-written manuscript and covers recent extensive reports in the field. In this manuscript, the author has covered extensive and comprehensive studies on EZH2 which is overexpressed in MPM and believed to be an oncogenic driver also in MPM. The further author also discussed about EZH2 as a potential diagnostic and therapeutic target. The author raises the scope for EZH2 inhibitors as a potential treatment option for MPM patients. This manuscript will be useful for readers in the field and can be accepted for publication in its present form.

Author Response

Dear referee,

we would like to thank you for the time and the efforts you have spent for the evaluation of our manuscript. We are very grateful for your comments and we are glad to know that you enjoyed our review and found it interesting for scientist working the in the field of mesothelioma.

Best regards

Chiara Porta

Reviewer 2 Report

In this article, Al Khatib et al. review the benefits and challenges of inhibiting EZH2 in malignant pleural mesothelioma.

General comment:

The topic is of a great interest. The article is well written and logically subdivided into subsections.

Specific points:

In introductory part tazemetostat is mentioned, however, only later in text it is described in detail what tazemetostat is. I suggest to briefly mention in introductory part that tazemetostat is an EZH2 inhibitor.

Since cisplatin and pemetrexed are the only first-line therapy for MPM, in section '4. EZH2 as a promising therapeutic target for MPM' it could be briefly stated that there are attempts in preclinical studies in other cancer types to use the EZH2 inhibition and cisplatin as a combination anticancer therapy.

The following sentence 'However, H3K27 demethylation observed upon DZNep treatment is due to proteolytic degradation of EZH2 and other PRC2 components, rather than specific EZH2 catalytic inhibition' is important for the comprehension of the mechanisms of DZNep action. Therefore, it would be good to put a reference at the end of this sentence.

Figure 1 could also benefit from the addition of references; either on the figure directly, or in the figure captions.

Author Response

Dear referee,

we are very grateful for having evaluated our review and raised fruitful comments to improve it. Below you can find our point-by-point reply to your comments (underlined text).

In introductory part tazemetostat is mentioned, however, only later in text it is described in detail what tazemetostat is. I suggest to briefly mention in introductory part that tazemetostat is an EZH2 inhibitor.

Thanks for the observation. We have now briefly described that tazemetostat is an EZH2 inhibitor (please see the underlined text at page 3)

Since cisplatin and pemetrexed are the only first-line therapy for MPM, in section '4. EZH2 as a promising therapeutic target for MPM' it could be briefly stated that there are attempts in preclinical studies in other cancer types to use the EZH2 inhibition and cisplatin as a combination anticancer therapy.

This is definitely a good point that we have now briefly discussed at the end of the subchapter 4.

The following sentence 'However, H3K27 demethylation observed upon DZNep treatment is due to proteolytic degradation of EZH2 and other PRC2 components, rather than specific EZH2 catalytic inhibition' is important for the comprehension of the mechanisms of DZNep action. Therefore, it would be good to put a reference at the end of this sentence.

Thanks for the observation, we have now included the missed reference

Figure 1 could also benefit from the addition of references; either on the figure directly, or in the figure captions.

Thanks again for the suggestion. Accordingly, we have now added the references on the figure directly

Reviewer 3 Report

I am really impressed by how well-written this review is. The authors have listed sound arguments for all the topics they discuss, the points they make are very well presented and the manuscript is so easy to read, in spite of its length and complexity. I will use this review as a great example of synthetic reasoning and critical analysis of the literature. I read through the text with great interest and I only stumbled at the beginning of the second section. The phrase "genetic mutational landscape" is redundant, mutational landscape is enough. I am also confused about how the molecular heterogeneity observed by the genetic variation indicates a "core" unique genetic signature. I think the authors need to restructure their argument to make their point more clear.   I also have 2 questions the authors could make a comment on if they think they are relevant. 1. Are there any clinical studies on MPM examining the potential interplay between EZH2 and UTX in the regulation of H3K27me3? Most of the interpretations of the role of EZH2 overexpression in MPM are through the deposition of its repressive histone mark. As an epigenetics expert, I always think of the levels of H3K27me3 (or any histone modification of interest) as the fine balance between the function of the methyltransferases and the demethylases involved in each case. 2. My second question is what is the physiological role of EZH2 in pleural mesothelial cells? I propose the acceptance of this manuscript with minor revision and editing of a few typos I noticed throughout.

Author Response

Dear referee,

We are very grateful and honored for the comments you raised about our review. We appreciate your valuable questions, which absolutely raised important points that we have now discussed, improving the quality of the manuscript. Below you can find our point-by-point reply to your comments (underlined text).

The phrase "genetic mutational landscape" is redundant, mutational landscape is enough. I am also confused about how the molecular heterogeneity observed by the genetic variation indicates a "core" unique genetic signature. I think the authors need to restructure their argument to make their point more clear.  

Thanks for the observation, we have now eliminated redundant terms and re-phrased the sentence in order to explain more clearly and properly the existence, beyond heterogeneity, of a limited number of genes that, since commonly mutated in almost all MPM, likely represent key “oncogenic drivers” in MPM.  The paragraph is below for convenience:

“The mutational landscape emerged over the last years has highlighted extensive genetic variation and gene expression deregulation both between and within MPM patients [17]–[20]. This molecular heterogeneity suggests the existence of a continuum of MPM clinical phenotypes, whose understanding will remarkably improve MPM classification and prognostication. In addition to the mutated genes that characterized each tumor,  most of MPM harbor loss of function mutations or genetic loss of few tumor suppressor genes (CDKN2A/2B, BAP1, NF2, TP53, LATS2 and SETD2), which likely play a key role in neoplastic transformation [17,18].”

 I also have 2 questions the authors could make a comment on if they think they are relevant. 1. Are there any clinical studies on MPM examining the potential interplay between EZH2 and UTX in the regulation of H3K27me3? Most of the interpretations of the role of EZH2 overexpression in MPM are through the deposition of its repressive histone mark. As an epigenetics expert, I always think of the levels of H3K27me3 (or any histone modification of interest) as the fine balance between the function of the methyltransferases and the demethylases involved in each case.

Given that the balanced activity of EZH2 methyltransferase with KDM6A (UTX), KDM6B (JMJD3) demethylases controls physiological H3K27me3 levels during development, we absolutely agree that both classes of enzymes deserve to be investigated to have a clearer and more relevant picture of the molecular mechanisms that regulate H3K27 methylation in MPM. Accordingly, accumulating amount of evidence has indicated that these proteins are linked with cancer cell features (e.g. proliferation, survival, stemness, migration, epithelial-mesenchymal transition) in different tumor types. However, Cregan and colleagues only have hitherto studied the impact of Kdm6 family members (Kdm6a and Kdm6b) on human MPM. At page 4, we have now briefly described the main findings of Cregan et al. and argued about the importance of getting insights that topic as perspective studies). The paragraph is below for convenience:

“Accordingly, accumulating amount of evidence has indicated that dysregulated activity of these proteins is linked with cancer cell features (e.g. proliferation, survival, stemness, migration, epithelial-mesenchymal transition) in different tumor types (doi:10.3390/cancers12102792). However, the interplay between EZH2 methyl-transferase and KDM6A (UTX) and KDM6B (JMJD3) demethylase has yet to be explored in MPM. The analysis of surgical samples from MPM patients showed that both KDM6A and KDM6B transcript levels are increased in malignant tumors (https://doi.org/10.3892/ijo.2017.3870). However, their pharmacological inhibition resulted in stronger anti-proliferative effects in normal mesothelial- than MPM- derived cell lines, reducing the interest in KDM proteins as therapeutic targets (https://doi.org/10.3892/ijo.2017.3870).”

  1. My second question is what is the physiological role of EZH2 in pleural mesothelial cells?

Few papers have studied the physiological role of EZH2 in pleural mesothelial cells. Morrisei and colleagues reported that Ezh2 is required for the proper differentiation of lung mesothelium by  suppressing smooth muscle differentiation. Ezh2 directly represses expression of the smooth muscle transcription factors, such as myocardin and Tbx18. At page 4, we have briefly mentioned this information about the role of EZH2 in physiological differentiation of lung mesothelium. We report the paragraph below for convenience:

“Albeit poorly studied in the context of pleural mesothelial cells (doi: 10.1242/dev.134932), it is widely recognized that the balanced activity of EZH2 methyltransferase with KDM6A (UTX) and KDM6B (JMJD3) demethylases controls physiological levels of H3K27me3, which drive proper cell differentiation during development”.

I propose the acceptance of this manuscript with minor revision and editing of a few typos I noticed throughout.

We apologize for typos, which we have now corrected.